# Cross-Bite and Oral-Health-Related Quality of Life (OHRQoL) in Preadolescents Aged 11 to 14 Years Old: A Pilot Case-Control Study

**DOI:** 10.3390/children10081311

**Published:** 2023-07-29

**Authors:** Adrián Curto, Alberto Albaladejo, Alfonso Alvarado-Lorenzo, Álvaro Zubizarreta-Macho, Daniel Curto

**Affiliations:** 1Department of Surgery, Faculty of Medicine, University of Salamanca, Alfonso X El Sabio Avenue s/n, 37007 Salamanca, Spain; albertoalbaladejo@usal.es (A.A.); kuki@usal.es (A.A.-L.); alvaro.zubizarreta@usal.es (Á.Z.-M.); 2Department of Pathology, 12 de Octubre University Hospital, Córdoba Avenue s/n, 28041 Madrid, Spain; daniel.curto@salud.madrid.org

**Keywords:** orthodontics, oral-health-related quality of life, malocclusion, cross-bite, preadolescent, self-concept

## Abstract

Introduction: Malocclusions have a negative impact on oral-health-related quality of life (OHRQoL). Posterior cross-bite is one of the most prevalent malocclusions in the preadolescent population. This study investigated the influence of posterior cross-bites (unilateral or bilateral) on OHRQoL in an 11- to 14-year-old population. Materials and Methods: A pilot case–control study was carried out at the Dental Clinic of the University of Salamanca between 2021 and 2023. A consecutive sample of 120 preadolescent patients aged 11 to 14 years old was recruited. Three groups were analyzed: a control group (no posterior cross-bite) (n = 40), a group with unilateral posterior cross-bite (n = 40), and a group with bilateral posterior cross-bite (n = 40). To analyze the OHRQoL, the Spanish version of the Child Perception Questionnaire (CPQ-Esp_11–14_) was used. Results: The mean age of the sample was 12.2 years old (±0.96 years). The group of patients with a bilateral posterior cross-bite was shown to have higher scores in all dimensions of the CPQ-Esp_11–14_, as well as a higher total score. Sex only influenced the oral symptom dimension of the CPQ-Esp_11–14_ questionnaire; in this dimension, the girls described a greater impact. Age did not influence OHRQoL. Conclusion: The presence of a posterior cross-bite had a negative impact on OHRQoL in the preadolescent population that was studied.

## 1. Introduction

Malocclusions are some of the most prevalent oral conditions in childhood. Of the different types of malocclusions, the posterior cross-bite is one of the most common [1]. The onset of a posterior cross-bite is usually seen in temporary and mixed dentition. The prevalence of posterior cross-bites ranges from 7.5% to 22% according to different authors [2,3,4]. This prevalence differs depending on the region analyzed. The prevalence of posterior cross-bites in children and adolescents is 4% on average in Europe and 17% on average in the Americas [5]. This prevalence is estimated to be higher in white populations than in Asian or African ethnic populations [6].

A posterior cross-bite is defined as an abnormal buccolingual relationship between the posterior teeth of the opposing arches. It can be caused by an individual malposition or a group malposition of posterior teeth. It can be classified as a unilateral or bilateral posterior cross-bite depending on the arches involved [7]. The state of occlusion in temporary dentition affects the development of occlusion in permanent dentition. The presence of a posterior cross-bite in temporary dentition can also imply its presence in permanent dentition. Most posterior cross-bites (50% to 90%) persist when the permanent teeth erupt [8].

There are multiple etiologies for posterior cross-bites, including dental crowding, temporary tooth ankylosis, premature loss of temporary teeth, habits such as thumb sucking, and even skeletal factors [9,10,11]. Recent studies that have analyzed the correlation between posterior cross-bites and the occurrence of temporomandibular joint pathology are inconclusive in this regard [12,13].

Regarding the treatment of cross-bites, it should be noted that self-correction only occurs in a small percentage of cases; therefore, early treatment is recommended when a posterior cross-bite is established. This treatment may involve expansion of the maxilla, elimination of any occlusal interference, or elimination of functional jaw displacement depending on the etiology of the posterior cross-bite [14].

The term used to assess patients’ perceptions of their psychosocial wellbeing and oral health is oral-health-related quality of life (OHRQoL). It is important to pay attention to early diagnosis and preventive treatment of malocclusions in children [15]. The presence of malocclusion affects a patient’s appearance, oral function, social life, and self-esteem; these previous points are part of the different components of OHRQoL [16]. It can be deduced that the self-perception that a patient has about their malocclusion will not always be directly related to the severity of the said malocclusion. It is essential to take the perspectives of patients into account when planning an orthodontic treatment [17]. It is also important to take into account the perceptions of the parents and/or caregivers of minors regarding the negative impacts that malocclusions have on the OHRQoL of the patients [18,19].

The Child Oral Health-Related Quality of Life (COHQoL) is a set of multidimensional scales that measure the negative effects that oral and orofacial diseases and disorders can have on the well-being of children aged 6–14 years and their families [20,21]. There are different questionnaires to assess COHQoL, each of which has been validated and used in patients of different ages; each questionnaire consists of a variable number of items. The most commonly used in published studies are the Child Perception Questionnaire (CPQ) [18], the Child Oral Impact Daily Performance (Child-OIDP) [22], the Child Oral Health Impact Profile (COHIP) [15,23] and the Early Childhood Oral Health Impact Scale (ECOHIS) [16,24,25].

There is a need to analyze the correlation between the presence of a malocclusion and the impact that the malocclusion has on the OHRQoL for children and preadolescents. In the scientific literature, there are differences among different authors. There are publications that have demonstrated a significant inverse relationship between malocclusion and OHRQoL [26,27,28]; however, others conclude that there were no statistically significant differences between these variables [29,30]. To the best of our knowledge, the impact of having a posterior cross-bite (both unilateral and bilateral ones) has not been analyzed in comparison with a control group with no cross-bite in order to evaluate the OHRQoL in preadolescent patients aged 11 to 14 years old.

The main objective of this study was to evaluate the impact of the presence of a posterior cross-bite (unilateral and bilateral) on the OHRQoL in a preadolescent population that were between 11 and 14 years of age by using the Spanish version of the Child Perception Questionnaire (CPQ-Esp_11–14_).

The null hypothesis of this study was that there were no statistically significant differences in OHRQoL between patients with a posterior cross-bite (unilateral or bilateral) and patients with an optimal occlusion.

## 2. Materials and Methods

### 2.1. Study Design

This pilot case–control study was carried out at the Dental Clinic of the University of Salamanca between 2021 and 2023. The protocol was approved by the Research Ethics Committee of the University of Salamanca (USAL 142/20). The study was carried out and reported in accordance with the STROBE guidelines [31].

The inclusion criteria for the control group meant that patients had to be aged between 11 and 14 years (both inclusive) with permanent dentition who did not need orthodontic treatment or needed mild orthodontic treatment (Orthodontic Treatment Need Index—Dental Health Component (IOTN-DHC) of 1 or 2) [32]. The inclusion criteria in the group of patients with unilateral posterior cross-bites were that they were patients with permanent dentition with an inverted occlusion relationship between at least one posterior tooth in the transverse plane in a hemiarch. The inclusion criteria in the group of patients with bilateral posterior cross-bites were that they were patients with permanent dentition with an inverted occlusion ratio between at least one posterior tooth in the transverse plane in both hemiarches [33].

The exclusion criteria for the three study groups were that the patients had mixed dentition, untreated dental caries, untreated gingival pathologies, previous orthodontic treatments, craniofacial syndromes, symptoms and/or diagnoses with temporomandibular joint pathologies, and chronic treatment with anti-inflammatory, analgesic, and/or anxiolytic drugs.

A consecutive sampling method was employed for convenience. The participants in this study were recruited before initiating any type of orthodontic treatment. The parents or legal guardians of the children signed a document consent form before the inclusion of the present study. The subjects were assured that all of their data would be confidential and that only the results of the study would be communicated. To ensure the anonymity of the patients, the name of each participant was replaced by a randomly assigned number. The age and sex of the study participants were also recorded to evaluate the influence of these two variables on OHRQoL.

The oral examinations were performed by a single clinical examiner to avoid errors and discrepancies between different examiners. The examiner was the main author of this research project, a dentist with clinical training and experience in paediatric dentistry and orthodontics.

### 2.2. OHRQoL Analysis

To evaluate the OHRQoL in the study participants, we used the Spanish version of the Child Perceptions Questionnaire (CPQ-Esp_11–14_), which was validated for children between 11 and 14 years of age [34]. The Child Perceptions Questionnaire was developed by Jokovic in 2002 [35]. This tool consists of 37 items that are grouped into four categories: oral symptoms, functional limitations, emotional wellbeing, and social wellbeing. The questions in this questionnaire refer to the frequency of events and feelings in the last three months.

The CPQ-Esp_11–14_ has a Likert scale structure, and the answer options are: “0 = Never”, “1 = Once/Twice”, “2 = Sometimes”, “3 = Often”, and “4 = Every Day/Almost Every Day” [28]. Higher scores indicate a worse OHRQoL [36]. Two short versions of CPQ-Esp_11–14_ have been developed: an 8-question version and a 16-question version [37].

The OHRQoL questionnaire was completed during clinical examination of the study participants. The participants were instructed to answer the questions without the support of their tutors.

### 2.3. Statistical Analysis

Statistical analysis was performed by using the IBM-SPSS Statistics software application (version 25; IBM, Armonk, NY, USA).

We report the obtained results as the mean values and standard deviations. This study used statistical hypothesis tests including the Chi-Square test, the Kolmogorov–Smirnov test, the Kruskal–Wallis test, and the Mann–Whitney test. The results of this study were considered significant when *p* < 0.05 and highly significant when *p* < 0.01.

## 3. Results

### 3.1. Description of the Sample

A total of 120 participants were recruited for the study. They were categorized into three groups of equal size (n = 40): two study groups (one group with unilateral posterior cross-bite and one group with bilateral posterior cross-bite) and one control group. There were no dropouts, all questionnaires were completely filled in, and none had to be excluded due to missing data.

The participants’ ages were distributed in the range between 11 and 14 years old, with a mean age of 12.2 years (±0.96 years). In relation to sex, there was a homogeneous distribution: 50% were female (n = 60) and 50% were male (n = 60). The distributions with respect to age were quite similar between the three groups, with there being no statistically significant differences (*p* > 0.05) in participant age. Therefore, we concluded that the three study groups were homogeneous with respect to age and sex (Table 1).

### 3.2. Descriptive Analysis of the Child Perceptions Questionnaire (CPQ-Esp_11–14_)

Using a Likert scale, the total scores from a questionnaire are usually calculated by using the method of summing the items; however, in this case, given that the dimensions had different numbers of items (6, 9, 9, and 13, respectively), it was considered more advisable to use the average response method to obtain scores that could be compared with each other (on the same 0–4-point scale as the items).

Based on the results that were obtained, we saw that the variables analyzed (the different dimensions from the OHRQoL in the CPQ-Esp_11–14_, as well as the total score) had high homogeneity with narrow ranges and very low standard deviations. The average values for these variables were all within a very small range—between 1.25 and 1.36 points—in all dimensions, with the average total score being 1.30 (±0.19).

In the CPQ-Esp_11–14_ questionnaire, the emotional wellbeing dimension had, on average, the most negative impact in the total sample studied (1.36 ± 0.22), whereas the social wellbeing dimension was the dimension with the lowest average score in the questionnaire (1.25 ± 0.19) (Table 2).

### 3.3. Comparison among Study Groups

By analyzing the impact of posterior cross-bites on OHRQoL in the sample that was studied using the Child Perceptions Questionnaire (CPQ-Esp_11–14_), we concluded that there were statistically significant differences between the three study groups in each of the questionnaire’s dimensions regarding the OHRQoL, as well as in the total score.

In all dimensions of the questionnaire (oral symptoms, functional limitations, emotional wellbeing, and social wellbeing), as well as in the total score, study participants with a bilateral posterior cross-bite reported a more negative OHRQoL than the participants in the control group, who were the ones who described the best OHRQoL.

In the bilateral posterior cross-bite group, the dimension of OHRQoL that was impacted the most was emotional wellbeing (1.59 ± 0.21), whereas the dimension of social wellbeing presented the lowest score (1.44 ± 0.18). In the unilateral posterior cross-bite group, the emotional wellbeing dimension was also the one that patients described most negatively (1.27 ± 0.14), whereas the social wellbeing dimension was the one that presented the lowest score in the CPQ-Esp_11–14_ questionnaire (1.17 ± 0.10). This trend was also observed in the control group for emotional wellbeing (1.22 ± 0.09) and social wellbeing (1.13 ± 0.11) (Table 3) (Figure 1).

### 3.4. Analysis of the Influence of Sex on OHRQoL

When analyzing the influence of sex on the OHRQoL in the sample, we concluded that there were no statistically significant differences, except in the oral symptom dimension of the CPQ-Esp_11–14_ questionnaire. In the oral symptom dimension, we observed statistically significant differences (*p* < 0.05) in relation to the influence of sex; in this dimension, girls described a more negative impact on their OHRQoL (1.36 ± 0.30) than boys did (1.24 ± 0.26). For the other dimensions of the questionnaire, as well as for the total score, we concluded that sex could not be considered an influential factor (Table 4).

### 3.5. Analysis of the Influence of Age on OHRQoL

To evaluate the influence of age on the OHRQoL of the participants in this study, we grouped them into two age groups: 11–12 years (n = 78) (65%) and 13–14 years (n = 42) (35%). The results showed that the mean values in both age groups were very similar to each other, and no statistically significant differences being observed (*p* < 0.05). Therefore, we can conclude that, in this study, age was not a factor that influenced the OHRQoL that patients described (Table 5).

## 4. Discussion

This research project assessed the impact of posterior cross-bites (unilateral or bilateral) on OHRQoL in a preadolescent population (aged 11–14 years old). This work was planned as a pilot case study (a group of patients with unilateral posterior cross-bites and a group of patients with bilateral posterior cross-bites) with controls.

The main objective of the present study was to analyse the impact of posterior crossbites on OHRQoL in a preadolescent population aged 11–14 years.

During childhood, OHRQoL can be affected by different demographic, social, and/or cultural factors [15,17,18,38]. In the scientific literature, there is evidence of the effect that the presence of malocclusion can have on the OHRQoL in the adult population; however, for the childhood population, the current scientific evidence is more limited [39,40,41,42,43,44]. It is important to analyze the impact that malocclusions have on OHRQoL in children and preadolescent patients due to the important changes in craniofacial growth and the development of occlusions that occur during this stage [15,16,17,18].

To analyze the OHRQoL in children, different questionnaires have been developed and validated. Different tools can be used to assessment OHRQoL in children, such as the Child Oral Health Impact Profile (COHIP), the Early Childhood Oral Health Impact Scale (ECOHIS), or the Child Perception Questionnaire (CPQ), which are the most representative and most widely used in published studies. There are differences among these tools. Some instruments focus on the severity of changes in the OHRQoL, whereas others focus on the frequency of oral problems on OHRQoL [45,46]. The Child Perception Questionnaire was used in this study because it is a questionnaire that is widely used in different published studies. The CPQ_11–14_ is the most commonly used tool to analyse OHRQoL in children. This questionnaire is a valid tool for measuring COHQoL and has been extensively validated in different countries [26,29,30,36,47,48,49,50]. In this study, the use of this questionnaire was considered appropriate for the sample studied.

Kallunki et al. analyzed the OHRQoL in a population between 8 and 10 years of age with a sample of 93 participants, differentiating among a group of patients with an excessive overjet, a group of patients with a unilateral posterior cross-bite, and a group of patients with a correct occlusion. The study used the Child Perceptions Questionnaire 8–10 (CPQ_8–10_). This study concluded that there were no statistically significant differences in OHRQoL due to the presence of a unilateral posterior cross-bite compared with patients with a normal occlusion [50]. The results described by this author are in contrast to the results described in the present study, where we observed that a posterior cross-bite significantly influenced the OHRQoL in a preadolescent population.

Our previous study also analyzed the OHRQoL in a sample of preadolescent patients using the CPQ-Esp_11–14_ questionnaire; however, this was in a sample of asthmatic patients. In the asthmatic population that was studied, it was concluded that neither age nor sex influenced OHRQoL. These previously reported results are similar to those reported in the present study [26]. When analyzing age in the present study, there was a slight trend in which older patients (group aged 13–14 years) provided higher scores in most dimensions of OHRQoL in the CPQ-Esp_11–14_ questionnaire than the younger patients did; however, this did not obtain statistical significance.

In 2022, Baskaradoss et al. analyzed the OHRQoL in a preadolescent population using the CPQ_11–14_ questionnaire. In the population in this study, the dimension with the greatest impact on the OHRQoL was the oral symptom dimension, whereas the social wellbeing dimension had the lowest score on the questionnaire. This study also reported that the social welfare dimension had the least impact on OHRQoL in the three study groups analyzed (control group, unilateral posterior cross-bite group, and bilateral posterior cross-bite group) [51].

In the present study, age was not found to have a statistically significant influence on the impact that malocclusion had on OHRQoL in the studied population according to the Child Perceptions Questionnaire. Regarding the influence of age on the OHRQoL, there are contradictory results in the literature. A systematic review by Alrashed et al. in 2021 concluded that the effects of malocclusion on OHRQoL were statistically significantly influenced by the age of the preadolescent patients [41].

The following aspects can be considered strengths of the present study: in this pilot study the sample size of 120 participants was considered a large sample size to give statistical validity to the results obtained; three well-defined and homogeneous study groups were established (the control group, the unilateral posterior cross-bite case group, and the bilateral posterior cross-bite case group); there was no loss of study participants; and only one examiner was used in the present study to eliminate variability errors between different examiners. In the three study groups present in our study, no statistically significant differences were observed with respect to the age and sex of the participants. Therefore, this aspect can increase the reliability of the results described here, since this analysis was carried out on a sample of the preadolescent population that was homogeneous with respect to these two variables.

One of the possible practical applications of the results of this study is the analysis of the benefits of early orthodontic care on OHRQoL in preadolescent populations. Assessing OHRQoL is interesting for orthodontic professionals to supplement clinical findings and to better understand the demand for orthodontic treatment. Knowing the impact that malocclusion has on OHRQoL in a population can provide a greater understanding of the consequences of malocclusion.

### 4.1. Limitations

One of the principal limitations of this study is that it was conducted on a population in a specific age range (patients between 11 and 14 years). Another possible limitation was that a longitudinal study was not developed. These limitations should be taken into account when extrapolating the results of this study to the entire population of children. The results of this study cannot be extrapolated to children with different types of malocclusions.

### 4.2. Recommendations

In future studies, it would be interesting to evaluate patients in an older age range, as well as to analyze the influence that orthodontic treatment of the malocclusion has on patients’ OHRQoL. Future studies should analyze how parents perceive the presence or lack of malocclusion, in addition to evaluating the possible influences of other environmental factors, such as economic status, the region in which the patients live, the parents’ education level, etc. It would also be interesting to analyze how the use of different orthodontic or orthopedic devices for the treatment of posterior cross-bites (unilateral or bilateral) affects the OHRQoL of preadolescent patients.

## 5. Conclusions

This study showed that preadolescent patients with bilateral posterior cross-bites reported more negative scores for all dimensions of OHRQoL and for the total score in the CPQ-Esp_11–14_ questionnaire than participants in the control group.

A posterior cross-bite (unilateral or bilateral) influenced the OHRQoL of the preadolescent patients analyzed in this study.

Sex can be considered an influential factor in the oral symptom dimension of the CPQ-Esp_11–14_ questionnaire (*p* < 0.05). Girls described a more negative impact on this dimension than boys did.

Age could not be considered an influencing factor on OHRQoL in this study.

## Figures and Tables

**Figure 1 children-10-01311-f001:**
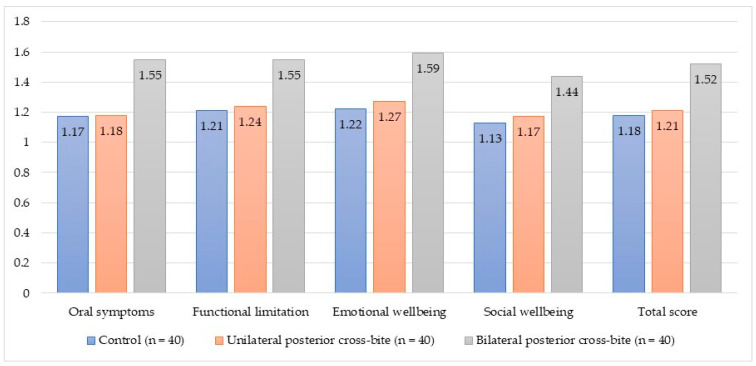
Assessment of oral-health-related quality of life in the different study groups (n = 120).

**Table 1 children-10-01311-t001:** Sociodemographic characteristics of the sample. Comparison among study groups (n = 120).

	Total Sample (n = 120)	Group			Chi-Square Test	
		Control (n = 40)	UPCB (n = 40)	BPCB (n = 40)	Statistical	*p*-Value
**Age**					2.17 ^ns^	0.904
11 years old	25.8% (n = 31)	25.0% (n = 10)	20.0% (n = 8)	32.5% (n = 13)		
12 years old	39.2% (n = 47)	42.5% (n = 17)	40.0% (n = 16)	35.0% (n = 14)		
13 years old	23.3% (n = 28)	22.5% (n = 9)	27.5% (n = 11)	20.0% (n = 8)		
14 years	11.7% (n = 14)	10.0% (n = 4)	12.5% (n = 5)	12.5% (n = 5)		
**Sex**					1.40 ^ns^	0.497
Female	50.0% (n = 60)	45.0% (n = 18)	47.5% (n = 19)	57.5% (n = 23)		
Male	50.0% (n = 60)	55.0% (n = 22)	52.5% (n = 21)	42.5% (n = 17)		

UPCB = Unilateral posterior cross-bite. BPCB = Bilateral posterior cross-bite. ns = Not significant (*p* > 0.05).

**Table 2 children-10-01311-t002:** Descriptive and exploratory analysis of the variables of the Child Perceptions Questionnaire (CPQ-Esp_11–14_) in the total sample (n = 120).

Categories	Centrality		Variability			Form		Kolmogorov–Smirnov Test *p*-Value
	Mean [95% CI]	Medium	Range	Standard Deviation	Interquartile Range	Skewness	Kurtosis	
**Oral symptoms**	1.30 (1.25–1.35)	1.33	0.83/2.17	±0.29	0.33	0.97	0.50	0.000
**Functional limitations**	1.33 (1.29–1.38)	1.33	0.89/2.00	±0.24	0.33	0.62	−0.09	0.000
**Emotional wellbeing**	1.36 (1.32–1.40)	1.33	0.89/2.00	±0.22	0.22	0.81	0.19	0.000
**Social wellbeing**	1.25 (1.21–1.28)	1.23	0.85/1.77	±0.19	0.15	0.56	−0.14	0.000
**Total score**	1.30 (1.27–1.34)	1.22	0.97/1.73	±0.19	0.27	0.78	−0.48	0

**Table 3 children-10-01311-t003:** Comparative inferential analysis of the variables in the oral-health-related quality of life questionnaire according to study group (n = 120).

Categories	Mean (±Standard Deviation)			Kruskal–Wallis Test		Effect Size: R^2^
	Control (n = 40)	UPCB (n = 40)	BPCB (n = 40)	Statistical	*p*-Value	
**Oral symptoms**	1.17 (±0.17)	1.18 (±0.16)	1.55 (±0.31)	38.94 **	0.000	0.384
**Functional limitation**	1.21 (±0.14)	1.24 (±0.14)	1.55 (±0.24)	44.73 **	0.000	0.426
**Emotional wellbeing**	1.22 (±0.09)	1.27 (±0.14)	1.59 (±0.21)	59.60 **	0.000	0.529
**Social wellbeing**	1.13 (±0.11)	1.17 (±0.10)	1.44 (±0.18)	54.42 **	0.000	0.515
**Total score**	1.18 (±0.08)	1.21 (±0.09)	1.52 (±0.17)	63.46 **	0.000	0.629

UPCB = Unilateral posterior cross-bite. BPCB = Bilateral posterior cross-bite. ** = Highly significant (*p* < 0.01).

**Table 4 children-10-01311-t004:** Comparative inferential analysis of the variables in the Child Perceptions Questionnaire (CPQ-Esp_11–14_) by sex (n = 120).

Categories	Mean (±Standard Deviation)		Mann–Whitney U Test		Effect Size: R^2^
	Boys (n = 60)	Girls (n = 60)	Statistical	*p*-Value	
**Oral symptoms**	1.24 (±0.26)	1.36 (±0.30)	2.37 *	0.018	0.042
**Functional limitations**	1.32 (±0.23)	1.35 (±0.25)	0.59 ^ns^	0.556	0.003
**Emotional wellbeing**	1.35 (±0.22)	1.37 (±0.23)	0.42 ^ns^	0.675	0.003
**Social wellbeing**	1.23 (±0.17)	1.26 (±0.21)	0.17 ^ns^	0.685	0.004
**Total score**	1.28 (±0.18)	1.32 (±0.21)	0.92 ^ns^	0.355	0.011

ns = Not significant (*p* > 0.05). * = Significant (*p* < 0.05).

**Table 5 children-10-01311-t005:** Comparative inferential analysis of the variables in the Child Perceptions Questionnaire (CPQ-Esp_11–14_) by age (n = 120).

Categories	Mean (±Standard Deviation)		Mann–Whitney U Test		Effect Size: R^2^
	11–12 Years (n = 78)	13–14 Years (n = 42)	Statistical	*p*-Value	
**Oral symptoms**	1.31 (±0.29)	1.27 (±0.28)	1.07 ^ns^	0.286	0.007
**Functional limitations**	1.32 (±0.22)	1.35 (±0.26)	0.14 ^ns^	0.891	0.002
**Emotional wellbeing**	1.35 (±0.21)	1.38 (±0.25)	0.40 ^ns^	0.687	0.005
**Social wellbeing**	1.24 (±0.19)	1.26 (±0.20)	0.48 ^ns^	0.632	0.004
**Total score**	1.30 (±0.18)	1.31 (±0.21)	0.54 ^ns^	0.589	0.011

ns = Not significant (*p* > 0.05).

## Data Availability

The data presented in this manuscript are available on request from the corresponding author.

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
