# Peer review of "Cross-Bite and Oral-Health-Related Quality of Life (OHRQoL) in Preadolescents Aged 11 to 14 Years Old: A Pilot Case-Control Study"

_children, 2023, doi:10.3390/children10081311_

Round 1

Reviewer 1 Report

Overall comments:

Very well-done study comparing the oral health quality of life for patients with unilateral, bilateral posterior cross bite malocclusion with normal occlusion among 11-14 years old. No major flaws. Conclusion seems to be supported by the evidence provided. Paper just needs minor corrections. Please refer to the below comments for necessary corrections.

Specific comments:

Methods:

2.1 Study design: Please provide more information on how was the sample size of 120 determined

Please provide more information on how many providers were involved in clinical examination and data collection? What training did they get? Any precautions taken to avoid any provider bias?

Results:

Table 1: Please specify the statistical tests used for the p-value to compare the differences between the groups.

It would be interesting to see if the difference between unilateral and bilateral cross bites is statistically significant. So, if available please share the results of pair-wise comparisons between the groups.

Would recommend including a graph especially for table 3, so that it is easier for reader to follow the trends between groups.  

References:

The following recent article on OHQoL and malocclusion could be used as additional reference –

Baskaradoss, J.K., Geevarghese, A., Alsaadi, W. et al. The impact of malocclusion on the oral health related quality of life of 11–14-year-old children. BMC Pediatr 22, 91 (2022). https://doi.org/10.1186/s12887-022-03127-2

Reviewer 2 Report

Dear Authors, this paper about Cross bite and oral health related quality of life in patients between 11 and 14 years old is really interesting and well performed. I am sure that this study will be helpful for both scientists and dental workers. 

Some issues need to be solved before its final publication in the journal.

Abstract: please divide the abstract into "introduction", "materials and methods", "results", "conclusion"

Introduction: This is a really important part of an article, especially in this one, this part is too short. The Authors should better describe the "Oral health related quality of Life" questionnaire. Some already published paper could help improve this introduction:

-  Ludovichetti FS, Zuccon A, Cantatore D, Zambon G, Girotto L, Lucchi P, Stellini E, Mazzoleni S. Early Childhood Caries and Oral Health-Related Quality of Life: Evaluation of the Effectiveness of Single-Session Therapy Under General Anesthesia. Eur J Dent. 2022 Oct 28. doi: 10.1055/s-0042-1757210.

Raghu R, Gauba K, Goyal A, Kapur A, Gupta A, Singh SK. Oral Health-related Quality of Life of Children with Early Childhood Caries before and after Receiving Complete Oral Rehabilitation under General Anesthesia. Int J Clin Pediatr Dent. 2021;14(Suppl 2):S117-S123.

Materials and methods and results are well written and easy to understand.

Discussion: 

  1. The discussion lacks a clear and specific research objective. It should explicitly state the main purpose of the study, such as investigating the impact of posterior cross-bites on the oral health-related quality of life (OHRQoL) in preadolescent populations. This will help readers understand the significance of the research and its contribution to the existing body of knowledge.

  2. The discussion mentions the use of the Child Perception Questionnaire (CPQ) to analyze OHRQoL in children. However, it does not provide a sufficient rationale for choosing this particular questionnaire over other available options, such as the Child Oral Health Impact Profile (COHIP) or the Early Childhood Oral Health Impact Scale (ECOHIS). Justifying the selection of CPQ in the context of the study's objectives and its advantages over other tools would enhance the credibility of the research.

  3. The study's limitations section highlights the absence of a longitudinal study design. To strengthen the research findings and draw more robust conclusions, it is essential to consider conducting longitudinal studies that follow participants over an extended period. This will provide valuable insights into the long-term impact of posterior cross-bites on OHRQoL and help establish causality between malocclusion and quality of life outcomes.

Additionally, while not explicitly mentioned in the discussion, it would be beneficial to address the issue of the sample size and diversity of the study participants. Ensuring a representative and diverse sample can improve the generalizability of the study findings to the broader population of preadolescent children.
